# Multi-Omics Analysis Revealed the rSNPs Potentially Involved in T2DM Pathogenic Mechanism and Metformin Response

**DOI:** 10.3390/ijms25179297

**Published:** 2024-08-27

**Authors:** Igor S. Damarov, Elena E. Korbolina, Elena Y. Rykova, Tatiana I. Merkulova

**Affiliations:** 1Institute of Cytology and Genetics, Siberian Branch of Russian Academy of Sciences, 630090 Novosibirsk, Russia; damarovis@bionet.nsc.ru (I.S.D.); rykova.elena.2014@gmail.com (E.Y.R.); merkulova@bionet.nsc.ru (T.I.M.); 2Department of Engineering Problems of Ecology, Novosibirsk State Technical University, 630087 Novosibirsk, Russia

**Keywords:** regulatory SNPs, allele-specific events, gene expression regulation, type 2 diabetes, metformin

## Abstract

The goal of our study was to identify and assess the functionally significant SNPs with potentially important roles in the development of type 2 diabetes mellitus (T2DM) and/or their effect on individual response to antihyperglycemic medication with metformin. We applied a bioinformatics approach to identify the regulatory SNPs (rSNPs) associated with allele-asymmetric binding and expression events in our paired ChIP-seq and RNA-seq data for peripheral blood mononuclear cells (PBMCs) of nine healthy individuals. The rSNP outcomes were analyzed using public data from the GWAS (Genome-Wide Association Studies) and Genotype-Tissue Expression (GTEx). The differentially expressed genes (DEGs) between healthy and T2DM individuals (GSE221521), including metformin responders and non-responders (GSE153315), were searched for in GEO RNA-seq data. The DEGs harboring rSNPs were analyzed using the Gene Ontology (GO) and Kyoto Encyclopedia of Genes and Genomes (KEGG). We identified 14,796 rSNPs in the promoters of 5132 genes of human PBMCs. We found 4280 rSNPs to associate with both phenotypic traits (GWAS) and expression quantitative trait loci (eQTLs) from GTEx. Between T2DM patients and controls, 3810 rSNPs were detected in the promoters of 1284 DEGs. Based on the protein-protein interaction (PPI) network, we identified 31 upregulated hub genes, including the genes involved in inflammation, obesity, and insulin resistance. The top-ranked 10 enriched KEGG pathways for these hubs included insulin, AMPK, and FoxO signaling pathways. Between metformin responders and non-responders, 367 rSNPs were found in the promoters of 131 DEGs. Genes encoding transcription factors and transcription regulators were the most widely represented group and many were shown to be involved in the T2DM pathogenesis. We have formed a list of human rSNPs that add functional interpretation to the T2DM-association signals identified in GWAS. The results suggest candidate causal regulatory variants for T2DM, with strong enrichment in the pathways related to glucose metabolism, inflammation, and the effects of metformin.

## 1. Introduction

Type 2 diabetes mellitus (T2DM) is the most common form of diabetes and one of the most prevalent metabolic disorders, which affects hundreds of millions of individuals worldwide [1]. The etiology of this multifaceted disease involves an interplay of multiple genetic, epigenetic, developmental, and environmental factors, including age, metabolic syndrome, and lifestyle. T2DM also overlaps in epidemiology, pathogenesis, and genetics as revealed by genome-wide association studies (GWAS) [2] with obesity; notably, there is also a strong immune connection between the two diseases [3,4].

The characteristics of T2DM are the alterations in circulating glucose level regulation, chronic hyperglycemia, and insulin resistance [5,6]. Emerging data suggest that the glucotoxicity caused by chronic hyperglycemia injures many cell types, such as pancreatic cells, leading to loss in functional β-cells [7]; hepatic cells, inducing endoplasmic reticulum stress; hepatocyte cell death; nonalcoholic fatty liver disease [8]; innate immune cells, inducing an inflammatory response [9]; and activation of monocytic lineage cells [10]. In turn, the chronic inflammatory state associated with the onset of T2DM leads to the development of long-term consequences: macrovascular complications, including a range of cardiovascular diseases [11,12] and microvascular complications, including retinopathy, nephropathy, and neuropathy [13]. Furthermore, it may contribute to the chronic inflammation of the central nervous system and neurodegeneration [14], as well as contribute to the association of T2DM with other conditions promoted by inflammation, such as rheumatoid arthritis [15].

The candidate genes [16,17,18] and GWAS [19,20] have linked single nucleotide polymorphisms (SNPs) at more than 250 loci in the human genome to T2DM risk, laying the foundations for functional investigations. Among them, the association of *TCF7L2* (transcription factor 7-like 2) with T2DM seems to be the most promising since many *TCF7L2* variants that increase the disease risk have been replicated in numerous studies and populations with diverse genetic origins [21]. The gene encodes a downstream effector of the canonical Wnt/β-catenin signaling pathway, which has been associated with several fundamental processes, including adipogenesis [22]. Most of the *TCF7L2* variants revealed by GWAS are located in noncoding genome regions suggesting that they exert their effects by modulating expression. The high-performance sequencing of formaldehyde-assisted isolation of regulatory elements (FAIRE-seq) in human islets demonstrated that *TCF7L2* at-risk variants were mapped in open chromatin sites. Among them, the heterozygotes of rs7903146 showed that enhancer activity depended on the allele [23]. A therapeutic potential was reported for some *TCF7L2* variants, for example, intronic rs290487 influenced the efficacy of repaglinide, an oral hypoglycemic agent, in Chinese T2DM patients [24]. 

Yet deciphering the functionality of the known risk variants on a genome-wide scale is still challenging. That is why different omics approaches are ever more frequently used to study the functional significance of noncoding variants. In particular, Yan et al. [25] measured the transcription factor–DNA interactions using an ultra-high-throughput multiplex protein–DNA binding assay, termed SNP-SELEX (SNP evaluation by Systematic Evolution of Ligands by Exponential enrichment). The authors examined the in vitro binding of 270 human transcription factors to common sequence variants. The input DNA library contained more than 380,000 oligonucleotides originally designed to represent over 95,000 SNPs. They covered 110 SNPs linked to T2DM susceptibility by GWAS at the start of the project, 6724 SNPs in high LD with them (r^2^ ≥ 0.8), and 89,162 common SNPs in annotated candidate cis-regulatory sequences located within 500 kb of GWAS-derived SNPs. As a result, approximately 11% of the input SNPs (11,079 SNPs) showed significantly differential binding to at least one transcription factor to predict the potential impact on the involved molecular pathways.

In this study, we used a multi-omics approach to search for the functionally significant SNPs associated with T2DM. First, we obtained the ChIP-seq data for histone modifications, histone H3 lysine K4 trimethylation (H3K4me3), histone H3 lysine 27 acetylation (H3K27ac), and the transcriptome sequencing data (RNA-seq) for peripheral blood mononuclear cells (PBMCs) from nine healthy individuals. After the primary data processing and filtering, we searched for allele-asymmetric binding and expression events in paired ChIP-seq and RNA-seq data for each individual. Thus, we identified 14,796 rSNPs that influenced the expression of 5132 genes, of which about 40% coincided with the corresponding GWAS data. The top GWAS phenotypic associations for these rSNPs included T2DM and a group of closely related characteristics (waist–hip index, waist-to-hip ratio adjusted for BMI, and a body shape index). Then, we used publicly available RNA-seq data from both healthy individuals and the individuals with T2DM (GSE221521), including metformin responders and non-responders (GSE153315), to evaluate the rSNPs potentially involved in T2DM pathogenic mechanisms and drug response.

## 2. Results

### 2.1. Algorithm for Searching the rSNPs Associated with Allele-Specific Events in ChIP-seq and RNA-seq Data and Construction of rSNPs Panel

The rSNP search strategy comprised six main stages (Figure 1). At the first stage, PBMCs were isolated from the peripheral blood of nine healthy individuals. At the next stage, the active regulatory genome regions marked with histone modifications H3K4me3 and H3K27ac (ChIP-seq) were sequenced. Concurrently, the transcriptome of these PBMC samples was sequenced (RNA-seq). Allelic asymmetry was separately computed for each heterozygous SNP using the ChIP-seq and RNA-seq data. Next, allele-specific binding (ASB) SNPs located within ±1000 bp from the transcription start sites (TSSs) of the known genes were extracted from ChIP-seq data. At the final stage of the search for rSNPs, the promoter ASB SNPs were intersected with the corresponding genes with determined allele-specific expression (ASE) events. The resulting rSNPs panel was characterized using the GWAS [26,27], GTEx [28,29], and annotation and enrichment analysis of allele-specific transcription factor binding at SNP (ANANASTRA) [30,31] data. Then, the representation of rSNPs in the promoters of the genes with differential expression in the blood cells of T2DM patients versus healthy subjects (GSE221521) and in response to metformin (GSE153315) was assessed followed by the characterization of these genes utilizing the Search Tool for the Retrieval of Interacting Genes (STRING) [32,33], KEGG [34,35], GO [36,37], Reactome [38,39], and literature data.

The total volume of our NGS data is 278.3 GB (ChIP-seq, 142 GB and RNA-seq, 136.3 GB); on average, the information volume per individual is 15.35 GB (15.7 GB, ChIP-seq and 15.1 GB, RNA-seq), which is 177.6 million ChIP-seq paired-end reads and 176 million RNA-seq paired-end reads per individual. The determination of allele-asymmetric events allowed us to find 14,796 rSNPs associated with the expression of 5132 genes (Appendix A).

### 2.2. Characterizing Constructed rSNP Panel with GWAS, GTEx, and ANANASTRA

The interception of rSNPs with the genome regions within ±1000 bp from SNPs in the GWAS catalog (as of March 2023) demonstrated the association of 5688 rSNPs (38.4% of all rSNPs of the panel) with different phenotypic traits. Note that 1107 rSNPs (7.5% of all rSNPs of the panel) were directly represented in the GWAS catalog (Appendix A). The χ^2^ estimation demonstrated the rSNP panel enrichment in GWAS variants (*p*-value < 2.2 × 10^–16^; OR = 1.7; and 95% CI, 1.6–1.8) compared with all heterozygous non-regulatory SNPs.

The resulting rSNP set is enriched in GWAS phenotype associations (Appendix A) (Fisher’s test with Benjamini–Hochberg. p.adj < 0.1). Preliminarily, we excluded the traits associated with less than 100 GWAS variants from the analysis. In the resulting group of traits ranked according to the number of GWAS-derived rSNPs, most of the traits among the top 20 ones were associated with quantitative blood cell characteristics (Table 1). In addition, this list contains T2DM and the morphometric characteristics directly associated with this disease, namely, waist–hip index, waist-to-hip ratio adjusted for BMI, and a body shape index.

We searched the GTEx Consortium data as of March 2023 for the rSNPs associated with the altered gene expression in different tissues (search for eQTLs) and found that 9871 rSNPs (66.7% of the overall rSNP panel) are associated with eQTLs; notably, 2474 rSNPs were directly represented in the GTEx catalog and 7397 ones situated within ±1000 bp of them (Appendix A). Of all tissues represented in GTEx, the largest number of rSNPs, 2464, was expectedly detected in the whole blood (the data on individual blood cell fractions are absent in the catalog). The enrichment analysis demonstrated that the rSNP panel was enriched in the GTEx eQTL variants (*p*-value < 2.2 × 10^–16^; OR = 2.49; 95% CI, 2.38–2.6) as compared with all heterozygous non-regulatory SNPs identified in our ChIP-seq data.

In total, 4280 rSNPs (28.9% of all detected rSNPs) in 1628 genes were simultaneously associated with an eQTL effect (according to GTEx data) and a certain phenotypic trait (according to GWAS data) (Figure 2). Published data suggest the involvement of many of these genes in inflammatory processes (*ADAM17* [40,41,42], *AHNAK* [43,44], *AIF1* [45,46,47], *CCDC92* [48], *CTBP1* [49], *HCP5* [50], *MAPKBP1* [51], and *MAST3* [52]); obesity (*TP53INP1* [53], *TKT* [54], *ADAM17* [40,41], and *AHNAK* [55]), insulin resistance (*TP53INP1* [53], *TKT* [54], and *CCDC92* [48]); insulin secretion (*GIPR* [56], *KCNJ15* [57], and *AP3S2* [58], *ARAP1* [59]); proliferation of pancreatic β-cells (*CDKN1B* [60] and *CCND2* [61]); mitophagy (*TP53INP1* [53]); glycolysis (*TKT* [54]); and carbohydrate transport (*GBA2* [62]). Approximately 14% of these rSNPs (598) were associated with T2DM or tightly correlated with its traits according to GWAS data (Appendix A).

The intersection of our list of all 14,796 rSNPs with the variants detected in the ANANASTRA allele-specific binding loci of different transcription factors (ChIP-seq data) showed that 30.8% (4560, FDR < 0.05) of our rSNPs were localized to the allele-asymmetric binding sites of different transcription factors (Figure 2). The share of such rSNPs increases to 33.4% (1433, FDR < 0.05) for the list of 4280 rSNPs associated with eQTLs and different GWAS phenotypic traits. The largest number of the 4280 rSNPs was situated in the allele-specific binding sites of ANDR (*N* = 217), CTCF (*N* = 215), STAT1 (*N* = 167), BRD4 (*N* = 126), ESR1 (*N* = 121), YY1 (*N* = 97), GCR (*N* = 89), SPT5H (*N* = 76), STAG1 (*N* = 74), and ZFX (*N* = 63).

### 2.3. Analysis of Hub Genes, Modules, and Pathways in Associative Gene Networks of rSNP-Governed Differentially Expressed Genes Related to T2DM

#### 2.3.1. Search for Differentially Expressed Genes Related to T2DM and Harboring rSNPs within Promotors

As described above, the formed rSNP panel emerged to be enriched in the variants related to T2DM and the associated traits. To further clarify the putative role of the discovered rSNPs in the mechanisms underlying T2DM, we analyzed the RNA-seq (GSE221521) data deposited with the GEO datasets on the PBMCs of healthy subjects (*N* = 50), patients diagnosed with this disease (*N* = 74), and T2DM subjects with diabetic retinopathy (*N* = 69), representing the largest sample of the open access ones [63]. Note that these data were already searched for DEGs [63]; however, the authors did not give their full list. Correspondingly, we recomputed differential expression setting other significance thresholds (|log_2_FC| > 0.2 and p.adj. < 0.05).

We did not discover any DEGs when comparing the cohorts of healthy subjects and the T2DM patients without retinopathy. However, the comparison of the healthy cohort and the T2DM individuals with retinopathy allowed us to detect 4612 DEGs and 1284 of them contained 3810 rSNPs in their promoters. Moreover, 2481 rSNPs resided in the promoters of 772 upregulated DEGs (log_2_FC > 0) and 1329 rSNPs in the promoters of 512 downregulated DEGs (log_2_FC < 0) (Figure 3). Notably, among these rSNPs, rs893617 (*AP3S2*), rs1552224 (*ARAP1*), rs3744347 (*CBX1*), and rs2066827 (*CDKN1B*) are associated with T2DM according to GWAS data. The genes in which these rSNPs are located play an important role in the pathogenesis of T2DM [58,59,60]. As we mentioned earlier, *AP3S2* and *ARAP1* are involved in insulin secretion [58,59]. Increased expression of *CDKN1B* is associated with structural and functional changes in the kidneys in diabetic nephropathy [64], decreased pancreatic β-cell proliferation and serum insulin levels [60], as well as decreased macrophage proliferation and inflammation in atherosclerosis [65]. 

#### 2.3.2. Identification of Hub Genes and Analysis of Key Modules Using STRING-Based Protein Interactions, KEGG, and GO Enrichment

The detected DEGs carrying rSNPs in their promoters were searched for hub genes, typically defined as genes involved in the regulation of various biological processes via the interaction with their numerous target genes or proteins. We assumed that the rSNPs in the promoters of hub genes influenced their expression and thereby significantly interfered with the regulation of key processes associated with the development of diabetes and its complications.

Using STRING, we constructed a PPI network for the protein products of upregulated DEGs, comprising 735 nodes and 2243 edges. The computations of topological characteristics for each node in this network allowed us to identify the protein products of 31 hub genes (Appendix A). Hub genes were mainly represented by the genes involved in transcription regulation, namely, coding for transcription factors (*TP53*, *SREBF1*, and *RXRA*), a corepressor (*NCOR2*), a chromatin-binding protein (*BRD4*), histone deacetylase (*HDAC4*), and RNA polymerase II subunit RPB1 (*POLR2A*). The genes coding for serine/threonine kinases and membrane receptors formed a separate group. Many genes on this list are involved in different processes associated with T2DM development [66,67,68], such as obesity (*SREBF1* and *H6PD*), inflammation (*TGFB1*), and insulin resistance (*RPTOR* and *AKT1*). Notably, according to the analysis of the enrichment of hub genes in KEGG pathways, the insulin signaling pathway (*N* of genes = 7) displayed the maximum enrichment (p.adj. = 5.6 × 10^–7^). The top ten pathways with the highest enrichment values include AMPK (p.adj. = 3.93 × 10^–5^) and FoxO signaling pathways (p.adj. = 3.45 × 10^–6^), which are responsible for the regulation of glucose metabolism [69,70] (Figure 4).

Three significant modules were discovered in the PPI network constructed for upregulated genes. The most important of them (MCODE score = 5.514) comprised 36 nodes and 102 edges (Figure 5A) and included the protein products of nine hub genes: *TGFB1*, *RPTOR*, *HDAC4*, *BRD4*, *H6PD*, *NCOR2*, *RXRA*, *HTT*, and *ITGAM.* The analysis of enrichment in KEGG demonstrated that the genes coding for the proteins of this PPI module were involved in the PI3K-Akt signaling pathway (p.adj. = 1.98 × 10^−6^) and insulin signaling pathway (p.adj. = 1.75 × 10^−6^), directly associated with T2DM development [71]. Of the GO terms associated with T2DM pathogenesis, we identified those related to the immune response, namely, myeloid leukocyte activation (p.adj. = 0.003), inflammatory response (p.adj = 0.02), immune response (p.adj. = 0.04), mononuclear cell differentiation (p.adj. = 0.005), and leukocyte activation (p.adj. = 0.002) (Appendix A).

The PPI module second in significance comprises 33 nodes and 73 edges (Figure 5B) and includes the protein products of three hub genes: *TP53*, *POLR2A*, and *MDM2*. Analysis of the enrichment in GO terms showed that the genes in this module, similar to those in the first one, are involved in the innate immune response (p.adj. = 0.02). In addition, the response to peptides is among the significant terms (p.adj. = 0.04). A number of genes associated with this term (*TP53*, *MDM2*, *IRF5*, and *TYK2*) are directly involved in the insulin response [72,73,74,75]. The KEGG enrichment analysis failed to detect any statistically significant pathways associated with T2DM; however, such pathways were discovered using the Reactome data [38,39]: interferon signaling (p.adj. = 3.2 × 10^−6^), cytokine signaling in immune system (p.adj. = 1.8 × 10^−4^), and transcriptional regulation by TP53 (p.adj. = 0.005) (Appendix A).

The third module of 24 nodes and 50 edges (Figure 5C) comprises the protein products of five hub genes: *FASN*, *MAPK3*, *SREBF1*, *INSR*, and *NOTCH1.* KEGG pathway enrichment analysis discovered insulin resistance (p.adj. = 0.003), mTOR signaling (p.adj. = 0.003), T2DM (p.adj. = 0.03), insulin signaling (p.adj. = 0.003), and AMPK signaling (p.adj. = 0.01) pathways (Appendix A).

The constructed PPI network for the downregulated DEGs with the promoters harboring detected rSNPs comprises 485 nodes and 3961 edges. We distinguish 7 modules and 35 hub genes in this network, their protein products mainly belong to ribosomal subunits and chaperons (Appendix A). The analysis of enrichment in KEGG/GO/Reactome DEG protein products of each module detected the common terms and pathways related to ribosome biogenesis, transcription processes, rRNA processing, etc. (Appendix A), poorly interpretable with respect to T2DM pathogenesis. Presumably, the inhibition of protein synthesis is associated with the intake of antihyperglycemic drugs [76,77] and is a result of the therapy. The representation analysis demonstrates that the DEGs of this group are involved in the pathways involved in glucose metabolism, namely, insulin (*EIF4E2*) and PI3K/Akt signaling (*EIF4E2*, *GNB5*, *HSP90AB1*, *PPP2CA*, *CDKN1B*, *ATF4*, *BCL2*, and *YWHAB*) pathways. Moreover, *BCL2* and *HSP90AB1* were identified as hub genes.

Taking into account the upregulated genes, 15 DEGs were represented in the insulin signaling pathway and 22 DEGs, in the PI3K/Akt signaling pathway. Figure 6A,B show the scheme of the PI3K/Akt signaling pathway with highlighted up- and downregulated DEGs harboring rSNPs in their promoters.

#### 2.3.3. Selecting Important Regulators from PPI Network Using ROC Analysis

Using an additional ROC analysis stage, we distinguished the hub genes the expression level of which allowed the healthy individuals to be distinguished from T2DM subjects. Of 66 hub genes, 36 displayed AUC > 0.7, making them valuable T2DM predictors for the studied set of transcriptome data. The promoters of these hub genes contained 117 rSNPs which are putatively associated with T2DM development and are of great interest for further studies (Table 2).

### 2.4. Search for the rSNPs Potentially Associated with Individual Response to the Antidiabetic Drug Metformin

Using our rSNP panel, we searched for the rSNPs potentially associated with the response to metformin by analyzing the RNA-seq data (GSE153315, GEO DataSets) on the blood cells of T2DM non-responders (*N* = 10) and T2DM responders (*N* = 10) after a 3-month medication [78]. In total, 406 DEGs were found for these cohorts. As it emerged, the promoter regions of 131 DEGs harbored 367 rSNPs of our panel (Figure 7).

The group of genes coding for transcription factors emerged to be the largest among those that carried predicted rSNPs in their promoters, namely, (1) *FOXP1* (rs9827299 and rs927809457), coding for the well-known glucose homeostasis regulator [79], which is also a known suppressor of macrophage differentiation into an anti-inflammatory phenotype [80]; (2) *POU2F2* (rs3826705), the product of which is an important player in glucose homeostasis being an activator of the AKT/mTOR signaling pathway [81] and is involved in the cellular immune response by regulating B-cell proliferation and differentiation genes [82]; (3) *YY-1* (rs113799953 and rs760756349), which is expressed in all cell types being involved in a wide range of regulatory processes, including the regulation of lipid and glucose metabolisms and immune response [83]; (4) *KLF6* (rs3812715, rs17135808, rs3829201, rs10795076, and rs38127140), the product of which is associated with the differentiation of monocytes to macrophages [84]; (5) *NFIX* (rs11555274, rs1354007230, rs1293177918, and rs1235554616), the product of which is a putative myelopoiesis regulator [85]; and (6) *GMEB1* (rs1013370834, rs1022902336, and rs895890911), the product of which is necessary for a normal function of islet endothelial cells [86]. We also detected a considerable number of rSNPs in the promoter regions of the genes coding for the transcription factors with yet vague functions: *TSHZ2* (rs71354397), *ZNF121* (rs7253981), *ZNF44* (rs386901 and rs422563), *ZNF440* (rs1056484229), *ZNF611* (rs1350535255, rs8109324, rs11879101, rs707303, rs10419223, rs1044394923, and rs55769230), *ZNF714* (rs7258409, rs182211521, and rs61733856), *ZNF793* (rs2291003), and *ZNF813* (rs1467168591, rs1157700262, rs2015145, rs4599028, rs112581977, rs1968829, and rs181026484).

The genes of other transcription regulators are functionally related to this group. They include (1) *TAF3* (rs78092115 and rs2778473), coding for a basal transcription factor involved in the formation of RNA polymerase II initiation complex and histone modifications; the association of TAF3 locus and a decreased Mg^2+^ level in T2DM were recently demonstrated [87]; (2) *NCOR1* (rs8076864, rs1331834429, and rs178797), coding for a corepressor protein involved in insulin resistance [88]; (3) *SMARCA4* (rs2288844, rs17766161, rs1057226903, and rs4804550), coding for BRG1, a catalytic subunit of SWI/SNF chromatin remodeling complex involved in the regulation of inflammatory processes in many tissues; in particular, a deficiency in BRG1 decreases the risk of diabetic atherosclerosis in mice by suppressing an inflammatory response in vessels [89]; and (4) *PCBD2* (rs319597), the product of which is involved in the transcription upregulation by interacting with the transcription factor HNF1β, mutations in which are among the key genetic causes maturity-onset diabetes of the young [90].

Other distinguished groups of genes are involved in the transport function (*NPHP3*, *SEC62*, *SLC30A7*, *SLC35E4*, *SLC37A2*, *SPNS1*, and *USE1;* in total, they contain 13 rSNPs in their promoters); are membrane proteins (*SMIM14*, *SPAG1*, *TGOLN2*, *TMEM120B*, *TMEM245*, and *VAMP8;* in total, 14 rSNPs); are involved in ubiquitination (*NOSIP*, *RNF145*, *TRIM56*, *TRIM58*, and *UBE2G2;* in total, 12 rSNPs) and inflammation (*TNFSF14*, *PILRB*, *AIF1*, *PEBP1*, *CDE3;* in total, 11 rSNPs); and are long noncoding RNAs and RNA-binding proteins (*NSRP1*, *RBM26*, *SNHG3*, *SNHG5*, *TPT1-AS1*, and *DDX52;* in total, 12 rSNPs) and ribosomal proteins or the enzymes involved in ribosome assembly (*NSUN4*, *RPL14*, *RPL27A*, *RPL37A*, *RPLP2*, *RPS23*, and *RPS2P32;* in total, 24 rSNPs). Noteworthy, an analysis of the relevant literature demonstrated that several of these genes were involved in the processes underlying T2DM development/prevention and its complications [91,92,93,94,95,96,97,98,99,100,101,102].

## 3. Discussion

One of the main challenges in contemporary human genetics is the discovery of the genome variations influencing different biomedical traits and the elucidation of the molecular mechanisms allowing these variations to lead to phenotypic differences and complex diseases. So far, over half a million variants (mainly, SNPs) associated with various human traits and diseases have been discovered with the help of GWAS [27]. The overwhelming majority of these variants are situated in the noncoding part of the genome, which suggests their regulatory function [103,104]. Since the GWAS technology cannot give information about the functionality of the found variants, the research into a functional interpretation of the GWAS data at the level of individual SNPs and on a mass scale is increasing [103,105], as well as the design of the GWAS-independent large-scale functional approaches based on omics data analysis [29,103,106].

In this work, we used a multi-omics approach to search for the functional variants able to influence gene expression (rSNPs) in PBMCs, which are the most available human tissue involved in immunity and diabetes [107,108,109,110]. We detected allele-asymmetric events in our ChIP-seq data for H3K4me3 and H3K27ac histone marks and the transcriptome data (RNA-seq) for nine healthy individuals and identified 14,796 rSNPs in the promoter regions of 5132 genes. Note that obtaining similar data for other specific affected organs, such as pancreatic beta cells, liver, muscle, etc., would significantly expand the rSNP panel.

We discovered a considerable enrichment of our rSNP panel in GWAS variants as compared with a set of heterozygous non-regulatory SNPs (*p*-value = 2.8884 × 10^–8^), which confirmed a significant role of the discovered rSNPs in the formation of the traits. The intersection of rSNPs with the DNA regions within ±1000 bp of GWAS SNPs demonstrated the association of 5688 rSNPs (38.4% of all detected ones) with different phenotypic traits. Of them, 1107 rSNPs (7.5%) were directly represented in the GWAS catalog. This is a good matching of results for principally different approaches. On the one hand, this demonstrates that a considerable part of the detected rSNPs is actually associated with certain phenotypic traits and, on the other hand, uncovers the molecular sense of a large GWAS-identified SNP group.

We obtained a considerably better fit of results when comparing our 14,796 rSNPs to the GTEx project data on eQTL analysis, which is also a functional approach that links different SNP alleles to gene expression level [103]. It was found that 9871 rSNPs (66.7% of the panel) are associated with eQTLs; in addition, 2474 rSNPs were directly contained in the GTEx catalog and 7397 rSNPs fell within the ±1000 bp region from them. On the contrary, the result of the comparison of the functional approach utilizing the search for allele-asymmetric events in the binding sites of different transcription factors [31] was rather modest with only 39.4% (5844) of our rSNPs localized to the allele-asymmetric binding sites of different transcription factors. In our opinion, the better fit of our results to GTEx data compared with ANANASTRA is explainable with that our rSNP search pipeline is based on the analysis of both allele-specific binding (ASB) and ASE events of the corresponding genes, while ANANASTRA utilizes only ASB events.

The computation of the rSNP enrichment in GWAS variants (Table 1) shows the highest enrichment for the rSNPs associated with blood cell quantitative characteristics, which is expected because we used PBMCs. Among the diseases, the largest number of rSNPs were associated with T2DM, matching the published data on the involvement of mononuclear cells in the pathogenesis of this disease [111,112,113]. In addition, the rSNPs associated with several morphometric characteristics (body mass index, body shape index, and waist-to-hip ratio) were also identified; these rSNPs were mainly localized to the promoters of the genes involved in inflammatory processes [40,43,44,45], insulin resistance [53,54], obesity [40,41], and carbohydrate transport [62], suggesting their involvement on T2DM pathogenesis.

To further clarify the potential role of the rSNPs of our panel in the mechanisms of T2DM pathogenesis and its complications, we searched for rSNPs in the promoter regions of the genes the expression of which changed during T2DM development using the largest RNA-seq dataset of the GEO repository (GSE221521) for the PBMCs of healthy subjects and T2DM patients without and with diabetic retinopathy. Retinopathy is a T2DM complication occurring in approximately half of diabetes patients [114]. We did not discover any DEGs when comparing the healthy donors and the T2DM subjects without retinopathy. Presumably, this is associated with the effects of the therapy received by the patients (unfortunately, the authors of the paper did not indicate which particular therapy was used) and the selection of the controls, which included the individuals with HbA1c = 5.80%, typical to a prediabetic state according to certain data [115]. However, 4612 DEGs were identified when comparing the healthy cohort to the T2DM subject with retinopathy and 1284 of them contained 3810 rSNPs in their promoters. The PPI network for these DEGs demonstrated that the genes of key modules were involved in the insulin signal transduction pathways and immune response. In particular, *PIK3R2* and *PIK3CG*, represented in the first module code for the PI3K (phosphoinositide 3-kinase) subunit, an upstream regulator in the PI3K/AKT pathway. PI3K mediates the activation of the key enzyme in this pathway, Akt kinase (AKT serine/threonine kinase), promoting the glucose intake by cells via mediated activation of glucose transporters (GLUT) [116]; activation of glycogenesis via GSK-3 (glycogen synthase kinase 3) inhibition [117]; and negative regulation of gluconeogenesis by downregulating the expression of glucose-6-phosphatase [71] and phosphoenolpyruvate carboxykinase [118]. Note that *Akt1* is a key upregulated hub gene in the PPI network based on the highest values of topological parameters. The increase in *Akt1* expression in the PBMCs of the T2DM subjects as compared with healthy individuals was also reported by Manoel-Caetano et al. [119]. *TP53* (tumor protein p53), identified as the most significant hub gene, also plays an important role in inflammation and T2DM development. This gene codes for transcription factor p53, which as a rule inhibits inflammatory responses. In addition, p53 is able to directly bind to *Glut1* and *Glut4*, thereby directly inhibiting their transcription and promoting the development of insulin resistance. The upregulation of *TP53* expression is associated with a positive regulation of gluconeogenesis (via expression upregulation of glucose-6-phosphatase, phosphoenolpyruvate carboxylase, and so on) and a negative regulation of glycolysis (decreasing the levels of fructose-2,6-bisphosphate, phosphoglycerate mutase, glucose-6-phosphate dehydrogenase, and so on) [72]. Thus, our data suggest that the rSNPs influencing gene expression in the considered modules are most likely associated with T2DM development by interfering with glucose metabolism and immune response. As for the rSNPs in hub gene promoters, they can well be the key regulators of these processes and the alteration in their expression can influence their genes/target proteins as well. We distinguished 117 rSNPs harbored in hub gene promoters with a high potential of T2DM prediction in the analyzed dataset according to ROC analysis data. The rSNPs of this type are of the greatest interest for further studies.

We used the formed rSNP panel to analyze the RNA-seq data for the blood cells of T2DM metformin responders and non-responders, which allowed for new insight into the pharmacogenetic factors influencing the metformin response in this disease. Notably, the research into genetic foundations of individual susceptibility to drugs has so far mainly dealt with the search for variations in genes coding for phase I and II drug-metabolizing enzymes, drug transporters, and some upstream transcription regulators of known pharmacogenes [120,121,122]. However, a rather large pool of heterogeneous data on SNPs in the genes belonging to a drug-metabolizing system [123,124,125] sets the challenge to develop the approaches to a systematized search for such variants.

It is known that metformin, a first-line drug for T2DM, is ineffective for approximately 30% of the patients [126]. Association studies succeeded in detecting several tens of the SNPs associated with the individual response to metformin. Since metformin is not metabolized in the human body [127], these studies mainly focused on the genes coding for organic cation transporters (OCTs) and multi-antimicrobial extrusion (MATE) proteins. As a result, numerous SNPs associated with metformin pharmacokinetics and pharmacodynamics in the protein-coding and noncoding regions of several genes of these groups were discovered. These genes include *SLC22A1* and *SLC22A2*, coding for OKT1 and OKT2; *SLC47A1* and *SLC47A2*, coding for MATE1 and MATE2-K; *SLC2A*, coding for one of the key glucose transporters; and the gene encoding the SP1 transcription factor, which modulates the expression of metformin transporters [127,128,129,130,131]. GWAS supplemented this list with a number of genes the products of which perform other, first and foremost, regulatory functions. In particular, the variations associated with metformin response in the genes of the AMPK signaling pathway, namely, *STK11*, *PRKAA1*, and *PRKAA2*, were found [132]. The SNPs associated with the metformin response are also discovered near the *ATM* gene, the product of which is involved in the redox homeostasis in the cell, as well as in the *PRPF31* (pre-mRNA processing factor 31) gene; *CPA6* gene, coding for carboxypeptidase A6), involved in the regulation of Akt/mTOR signaling pathway; and *STAT3* gene, coding for the transcription factor actively involved in the regulation of metabolic and immune processes [133]. However, the use of GWAS in pharmacogenomics research requires a rather large cohort of patients with different responses to a drug. A tremendous number of currently used drugs and frequent cases of complex therapies considerably hinder obtaining mass data with the help of this approach [134,135]. An integrated analysis of the current omics functional data can help in resolving this problem, at least in part.

In this study, we, on the one hand, used our panel of 14,796 rSNPs constructed by detecting allele-specific events in ChIP-seq and RNA-seq data for the PBMCs of nine healthy individuals and, on the other hand, the transcriptome data (RNA-seq, GSE153315) for the blood cells of T2DM non-responders to metformin (after 3-month medication, *N* = 10) and the T2DM responders (*N* = 10) [78]. An integrated analysis of these data allowed us to obtain a new, wider list of the SNPs potentially associated with the individual response to metformin. Moreover, it has emerged that a considerable number of such SNPs reside in the promoter regions of the genes coding for transcription factors (*FOXP1*, *POU2F2*, *YY-1*, *KLF6*, *NFIX*, *GMEB1*, *TSHZ2*, *ZNF121*, *ZNF44; ZNF440*, *ZNF611*, *ZNF714*, *ZNF793*, and *ZNF813*) and other regulatory proteins (*TAF3*, *NCOR1*, *SMARCA4*, and *PCBD2*). The fact that rSNPs are discovered in these genes is important because of the possible modulation of an extensive reaction to the drug because these genes control different networks comprising numerous functionally linked genes. In particular, the YY-1 transcription factor is the best-studied of the mentioned transcription factors. It is ubiquitously expressed in many cell types and is able to activate or repress transcription of the networks comprising the genes involved in cell survival, replication, differentiation, metabolism, and inflammation [136,137]. The multifunctionality of YY-1 also appears as its involvement in the enhancer–promoter and super-enhancer–promoter interactions [138,139], splicing regulation [140], and transcription initiation by binding to the initiator element (INR, a core promoter) [141]. YY-1 has been shown to act in numerous processes the disturbance of which is associated with the development of diabetes and its complications as well as the regulation of systemic inflammation [83,142,143,144]. This suggests that the rSNPs discovered in the promoter-regulatory region of this gene are also potential contributors to the implementation of metformin therapeutic effects by altering its expression level and, as a consequence, the expression of YY-1 target genes. The assumption on an important role of YY-1 in the metformin response agrees well with the results of our analysis of ANANASTRA data [30,31] that the promoter regions of many genes differentially expressed in response to metformin carry the rSNPs influencing YY-1 binding.

## 4. Materials and Methods

### 4.1. Subjects of Investigation

The study involved nine somatically healthy individuals living on the territory of the Russian Federation (Novosibirsk, Russia) and displaying no diseases in their life history by the moment of sampling their biological material. The sample comprised males (*N* = 3) and females (*N* = 6) aged 26–42 (mean age, 33 years) (Appendix A). The study was approved by the ethical committee of the Institute of Therapy and Preventive Medicine, Novosibirsk, Russia. Each participant signed the informed consent.

### 4.2. Isolation of Peripheral Blood Mononuclear Cells

Blood (20 mL) was sampled from the cubital vein of each subject and diluted with an equal volume of phosphate-buffered saline (PBS). Ficoll (12 mL; ρ = 1.077, Biolot, Saint Petersburg, Russia) was added to two clean Falcon tubes to gently layer the diluted blood (20 mL per each Falcon tube) onto the Ficoll solution with a pipette. The samples were centrifuged at 400× *g* for 15 min at 4 °C. The layer of mononuclear cells from each Falcon was transferred to a new tube, supplemented with PBS (4 mL), mixed, and centrifuged (200× *g*/7 min/4 °C). The supernatant was removed and the pellet was supplemented with the buffer for lysing erythrocytes (20 mL; 150 mM NH_4_Cl and 0.01 mM EDTA) to incubate for 10 min at room temperature and centrifuge (200× *g*/7 min/4 °C). The supernatant was removed and the sediment was washed twice with PBS (3 mL) and centrifuged (200× *g*/7 min/4 °C). The purified cells from one Falcon tube were used to isolate RNA and from the other one, in the experiments on chromatin immunoprecipitation.

### 4.3. mRNA Sequencing

The purified cells obtained from each donor were homogenized in Trizol buffer (1 mL; Ambion, Austin, TX, USA), supplemented with chloroform (200 μL), and centrifuged (12,000× *g*/15 min/4 °C). The aqueous phase was transferred to a new tube, supplemented with 1/5 volume of chloroform, and centrifuged (12,000× *g*/5 min/4 °C) to transfer the supernatant into a new tube, precipitate it with one volume of isopropanol and 1 μL linear polyacrylamide (BIORON, Römerberg, Germany), and centrifuge (20,238× *g*/15 min/4 °C). The supernatant was collected to precipitate RNA with 75% alcohol (1 mL) and centrifuge (20,238× *g*/5 min/4 °C). The alcohol was removed and the pellet was dried at room temperature for 5–10 min to dissolve in 25 μL of ddH_2_O. RNA concentration was measured in a NanoDrop (Thermo Fisher Scientific, Waltham, MA, USA) spectrophotometer. The complementary DNA (cDNA) libraries were prepared using the NEBNext^®^ Ultra^™^ II Directional RNA Library Prep Kit for Illumina^®^ (New England Biolabs, Ipswich, MA, USA) according to the manufacturer’s protocol.

### 4.4. Chromatin Immunoprecipitation

The cell sediment was dissolved in PBS (5 mL) and 37% formaldehyde (138.8 μL; Sigma, Saint Louis, MO, USA), homogenized, and mixed in a rotator for 10 min at room temperature. The reaction was stopped with 2 M glycine (342 μL), mixed for 5 min in a rotator at room temperature, and centrifuged (200× *g*/5 min/4 °C); the supernatant was removed. The pellet was twice washed with PBS (3 mL) and supplemented with 800 μL of lysis buffer (10 mM Tris pH 7.5, 1 mM EDTA, 0.5% SDS, 0.1% sodium deoxycholate, and 1% Triton X-100) and 100X protease inhibitor cocktail (Thermo Fisher Scientific, Waltham, MA, USA) to break the chromatin into the fragments of 150–600 bp in an immersion homogenizer (BANDELIN electronic GmbH & Co. KG, Berlin, Germany). The resulting fragments were centrifuged (3000× *g*/10 min/4 °C) and the supernatant was added to A- and G-protein beads (20 μL; New England Biolabs, Ipswich, MA, USA) and supplemented with 1.5 V of the lysis buffer without SDS and 100X protease inhibitor cocktail. In parallel, G-protein beads (40 μL), lysis buffer (160 μL), lysis buffer without SDS (240 μL), and the antibodies (8 μL) to histone modification H3K4me3 or H3K27ac (Abcam, United Kingdom) were placed into a separate tube to incubate it together with the tube containing chromatin and A- and G-protein beads at a constant rotation for 2 h at 4 °C. The tubes were placed onto a magnetic base and then the supernatant was removed from the tube with antibodies and the sediment was supplemented with the supernatant after the incubation with A- and G-protein beads. The immunoprecipitation reaction was conducted overnight at +4 °C in a rotator. In the morning, the tube was returned onto magnetic base to remove the supernatant and washed twice with an addition of the lysis buffer (1 mL), 1 mL of high salt buffer (10 mM Tris pH 7.5, 1 mM EDTA, 0.1% SDS, 0.1% sodium deoxycholate, 1% Triton X-100, and 0.5 M NaCl), 1 mL of TE buffer (10 mM Tris pH 7.5, 1 mM EDTA, and 1% Triton X-100), and 1 mL of the TE buffer without Triton X-100. After removal of the supernatant, the tube was supplemented with 100 μL of elution buffer (1% SDS, 10 mM Tris pH 7.5, and 1 mM EDTA) and incubated for 1 h at +65 °C in a thermoshaker (800 rpm). The tube was placed onto a magnetic base and the supernatant was transferred into a new tube (the washing with elution buffer was repeated twice). To prevent chromatin cross-linking, the sample was supplemented with 5M NaCl (to a final concentration of 370 mM) and incubated in a thermostat at +65 °C for 4 h. The sample was supplemented with 2.46 volume of ddH_2_O, 0.5 M EDTA (1/50 of the sample volume), 1 M Tris-HCl pH 6.5 (1/16 of the sample volume), 10% SDS (1/13 of the sample volume) and 1 μL of proteinase K (800 U/mL; New England Biolabs, Ipswich, MA, USA) to incubate in a thermostat at +52 °C for 3 h. Then, RNase A (20 mg/mL; Thermo Fisher Scientific, Waltham, MA, USA) was added at the last 20 min. The sample was taken off the thermostat and supplemented with 1 volume of phenol–chloroform pH 8.0 (1:1) and centrifuged (1000× g/5 min) to transfer the aqueous phase to a new tube. Then, 1 volume of chloroform was added and centrifuged (1000× *g*/5 min). The aqueous phase was collected and supplemented with 1/10 V of 3 M NaAc, 1 volume of isopropanol, and 1 μL of linear polyacrylamide to incubate for 5–7 h at −70 °C followed by centrifugation (10,000× *g*/20 min/4 °C), removal of the supernatant, precipitation with 75% ethanol, and centrifugation (10,000× *g*/5 min/4 °C). The supernatant was removed and the pellet was dissolved in ddH_2_O (20 μL). DNA libraries were prepared analogously to the protocol for cDNA libraries.

### 4.5. Sequencing and Aligning to Reference Genome

Paired-end sequencing of cDNA and DNA libraries was performed using the NovaSeq 6000 or MGISEQ-2000RS platforms. The Illumina or DNBSEQ adapters and the reads shorter than 36 bp were discarded from the ChIP-seq and RNA-seq data with Trimmomatic v. 3.2.2 [145]. The resulting data were aligned to reference genome assembly GRCh38/hg38 (Genome Reference Consortium Human Build 38), available at the NCBI FTP site [146]. The forward and reverse reads (145–150 bp) were aligned with Bowtie 2 [147]; the threshold value of mapping quality (MAPQ) was >20. The aligned reads were transferred into bam format.

### 4.6. Search for Heterozygous Positions

To search for heterozygous positions, all ChIP-seq and RNA-seq files for each individual were pooled with the help of the bcftools merge command [148]. A position was regarded as heterozygous if it met the following conditions: coverage depth, >20 and frequency of minor allele, ≥0.2 of the total coverage of the corresponding position. The heterozygous positions were selected using bcftools mpileup [148].

### 4.7. Assembling Alternative Genome

In order to avoid the alignment bias towards the reads containing reference alleles (resulting from the mapping algorithms that discriminate the alternative allele), an additional individual alternative genome was assembled for each person; in this genome, the reference allele was replaced with the alternative one for each heterozygous position. Then, the ChIP-seq and RNA-seq file carrying the reads for each individual was aligned to the individual alternative sequence.

### 4.8. Computing Allelic Asymmetry

Allelic asymmetry for all heterozygous positions in the ChIP-seq and RNA-seq data was computed using the binomial test against the null hypothesis on an equal coverage of heterozygous positions by the reference reads aligned to reference genome and the alternative reads aligned to alternative genome with subsequent correction for multiple comparison according to Benjamini–Hochberg (p.adj. < 0.1). The rs ID numbers were assigned to the computed allele-asymmetric positions based on the dbSNP Build 155 [149,150]. Allele-asymmetric SNPs from ChIP-seq data were considered to be associated with allele-specific binding (ASB) events, and allele-asymmetric SNPs from RNA-seq data were considered to be associated with allele-specific expression (ASE) events.

### 4.9. Searching for Allele-Asymmetric SNPs in the Promoters of the Genes Displaying Allele-Specific Expression

The list of ASB SNPs for the ChIP-seq data was intersected with gene promoter regions. The region of ±1000 bp from the transcription start site of all transcripts (including alternative transcription start sites) contained in the TxDb.Hsapiens.UCSC.hg38.knownGene R annotation package for TxDb object(s) [151] was regarded as promoter. The list of ASE SNPs based on the RNA-seq data was intersected with the coordinates of the genes contained in the TxDb.Hsapiens.UCSC.hg38.knownGene package. The genes containing at least one ASE SNP were regarded as expressed in an allele-specific manner. The total allele-specific expression was computed for the genes carrying more than one ASE SNP using the MBASED approach [152]; this method is based on the merging of the information about allelic asymmetry for each heterozygous SNP in a particular gene. The final stage consisted of the intersection of the ASB SNPs localized to promoters and the target genes with the computed allele-specific expression. The promoter ASB SNPs, for which the target genes are expressed in an allele-specific manner, were named regulatory SNPs (rSNPs).

### 4.10. Characterizing rSNPs with the Help of Open-Access Data

The data of the GTEx project on eQTLs [28,153], GWAS Catalog on the association of genome loci and phenotypic traits [26,27], and ANANASTRA web server for annotation of the loci of allele-specific transcription factor binding in ChIP-seq data [30,31] were used to characterize the constructed rSNP panel.

### 4.11. Analyzing DEGs

DEGs were searched for using the DeSeq2 (version 1.44) R package [154] in the RNA-seq data deposited with the Gene Expression Omnibus database [155,156]. These data were obtained for the whole blood samples of the cohorts of (1) healthy subjects (*n* = 50) and T2DM cases with diagnosed (*n* = 69) and not diagnosed (*n* = 74) diabetic retinopathy (GSE221521) [63] and (2) healthy subjects (*n* = 10) and T2DM responders (*n* = 10)/non-responders (*n* = 10) to 3-month metformin treatment (GSE153315) [78]. According to [78] the patients who had a decrease in glycated hemoglobin (HbA1c) level > 1% or a decrease of >20 mg/dL in fasting blood glucose level from baseline after three months of therapy were considered as responders.

Multiple testing correction according to Benjamini–Hochberg procedure setting the false discovery rate (FDR) of <0.05 was used. Genes were regarded as differentially expressed if |log_2_FC| > 0.2 and p.adj. < 0.05. Then, DEGs were intersected with the genes harboring rSNPs from the constructed panel in their promoters.

### 4.12. Search for Hub Genes

The hub genes were searched for applying a network of protein interactions constructed with the help of STRING [32,33], which predicts protein–protein interactions (PPI) based on the data on physical (experimental data) and functional (computational predictions, co-expressions, and previous knowledge in databases) associations. The PPI network was separately constructed for the upregulated and downregulated DEG groups, which harbored rSNPs in their promoters. Then, three topological characteristics—degree, stress centrality, and betweenness centrality—were computed using CytoNCA [157,158], a Cytoscape plugin [159,160]. The lists of the top 20 proteins with the highest values of each computed characteristic were formed and merged. The proteins in the consolidated list were referred to as hub proteins and the corresponding genes, as hub genes.

### 4.13. Identification of Protein Modules

The significant modules in the analyzed PPI networks were identified using the MCODE tool [161,162]; this tool clusters the densely connected regions in large protein–protein interaction networks. The following criteria were used for selecting the significant modules: max depth = 100, degree cut-off = 2, node score cut-off = 0.2, MCODE scores ≥ 4, and K-score = 2.

### 4.14. Enrichment Analysis of KEGG/Reactome/GO

Functional enrichment analysis of all significant modules was performed. The terms from the GO Knowledgebase [36,37] together with the pathways from the KEGG [34,35] and Reactome Knowledgebase [38,39] were used as functional terms. The enrichment was analyzed using over-representation analysis (ORA) of the GO terms and KEGG/Reactome pathways in the protein modules with the R clusterProfiler package (version 4.12.5) [163]. After the Benjamini–Hochberg procedure, p.adj. < 0.05 was set as the enrichment threshold value.

### 4.15. ROC Analysis

ROC (Receiver Operator Characteristic) curves were constructed and AUC (Area Under Curve) values were computed from the normalized expression values for each hub gene calculated using DESeq2. The hub genes with AUC > 0.7 were regarded as good predictors of T2DM in the studied transcriptome dataset. The ROC curves were constructed using the pROC package (version 1.18.5) for R [164].

## 5. Conclusions

The search for allele-specific events in the experimental data on gene expression profiles (RNA-seq) and active chromatin marks H3K4me3 and H3K27ac (ChIP-seq) in the peripheral blood mononuclear cells of nine healthy donors allowed us to construct the panel comprising 14,848 rSNPs in the promoter regions of 5132 genes. Over 38% of the discovered rSNPs are associated with different GWAS phenotypic traits and 66.7%, with GTEx eQTLs, which uncovers the molecular sense of a large GWAS-identified SNP group and is an independent confirmation for the regulatory function of a considerable group of variants from GTEx catalogs. The rSNPs potentially associated with T2DM development and/or diabetic retinopathy as well as with individual specific features of the response to the antidiabetic drug metformin are identified. The genes carrying these rSNPs in their promoters are enriched in the pathways related to glucose metabolism and inflammation.

## Figures and Tables

**Figure 1 ijms-25-09297-f001:**
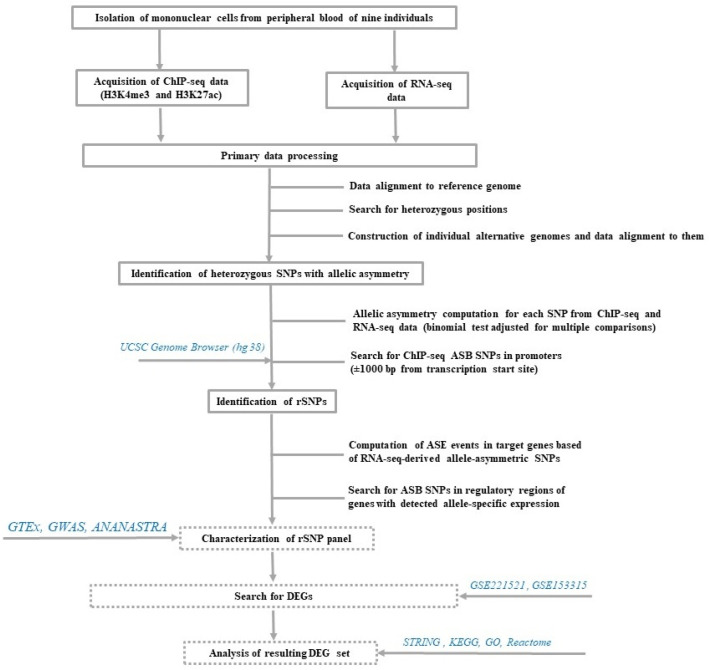
Scheme of the main stages in the search for rSNPs and their further analysis. The solid gray frame shows the stages of the search for rSNPs; the dotted gray frame, further analysis of the rSNP panel; and the cyan italic, data sources.

**Figure 2 ijms-25-09297-f002:**
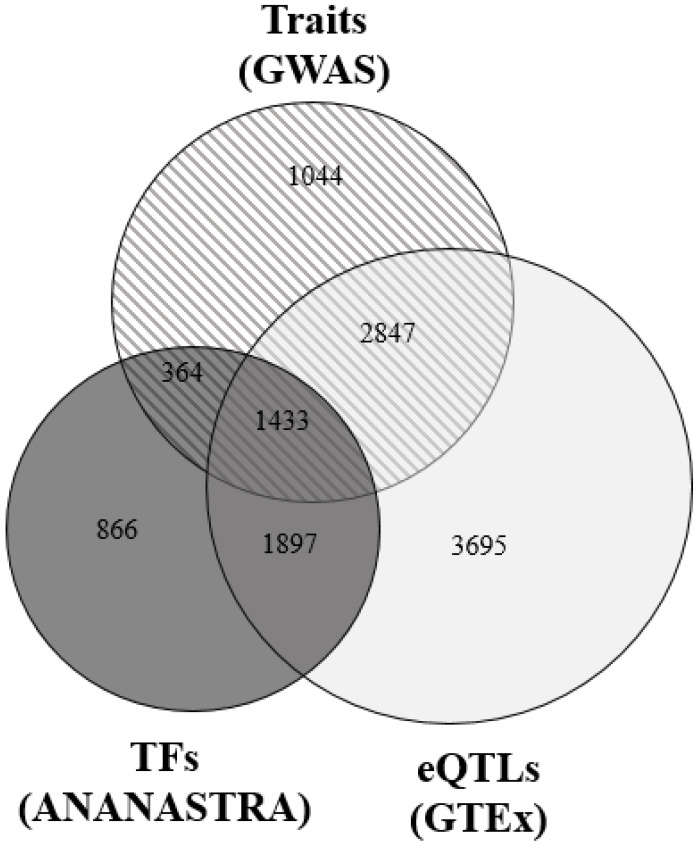
Venn diagram showing the number of the rSNPs localized to allele-specific transcription factor binding sites (ANANASTRA), rSNPs associated phenotypic traits (GWAS data), and eQTLs effects (GTEx data).

**Figure 3 ijms-25-09297-f003:**
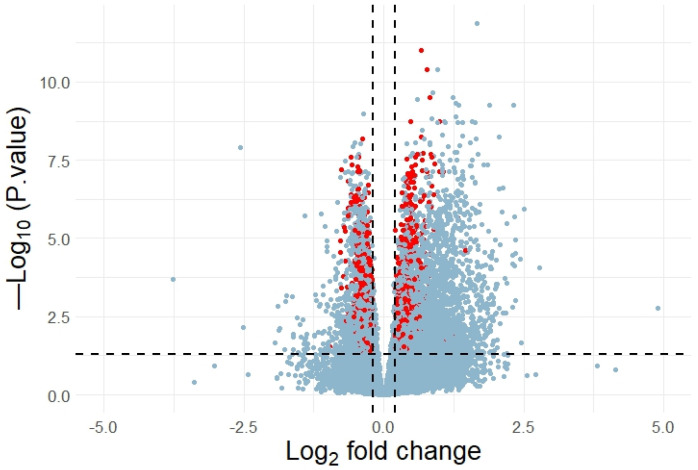
Volcano plot of DEGs. The horizontal axis stands for log2 fold change and the vertical axis, for –log_10_ (adjusted *p*-value). Statistically significant DEGs harboring rSNPs in their promoters are marked red.

**Figure 4 ijms-25-09297-f004:**
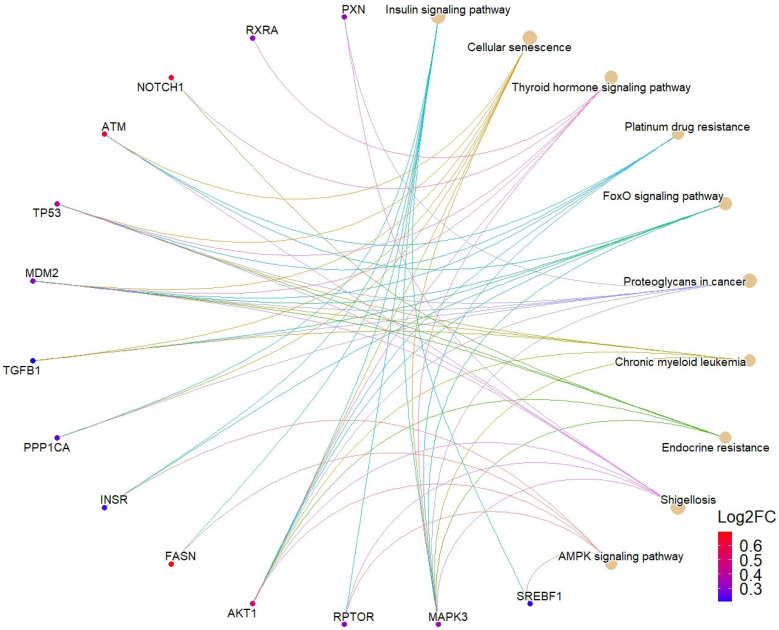
Network chart illustrating the link of top ten significant KEGG pathways according to the enrichment analysis with upregulated hub genes.

**Figure 5 ijms-25-09297-f005:**
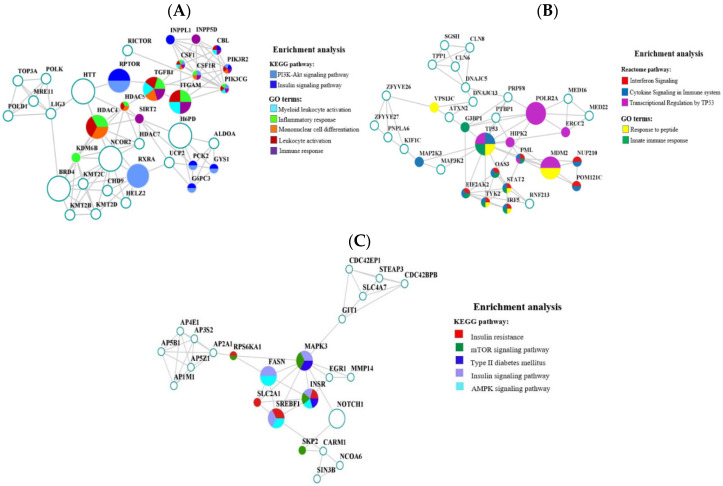
PPI network in the significant modules analyzed for KEGG and GO enrichment. (**A**) First module. (**B**) Second module. (**C**) Third module. In the PPI network, nodes show proteins and edges, their interaction. Hub proteins are denoted with larger symbols and the KEGG- and GO-annotated proteins, with the corresponding color (see the legend).

**Figure 6 ijms-25-09297-f006:**
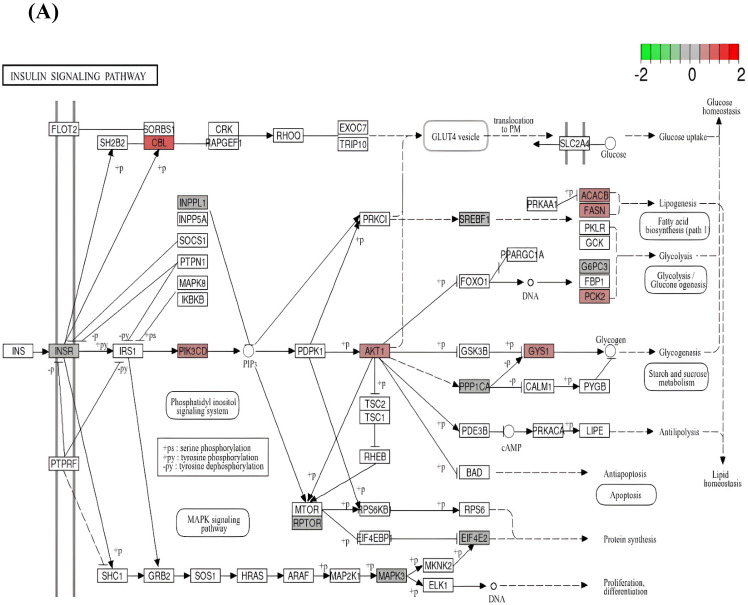
Graphical visualization of DEG representation in (**A**) insulin and (**B**) PI3K/Akt signaling pathways (KEGG data). Colors of nodes show the direction and value (log2FC) of expression alteration.

**Figure 7 ijms-25-09297-f007:**
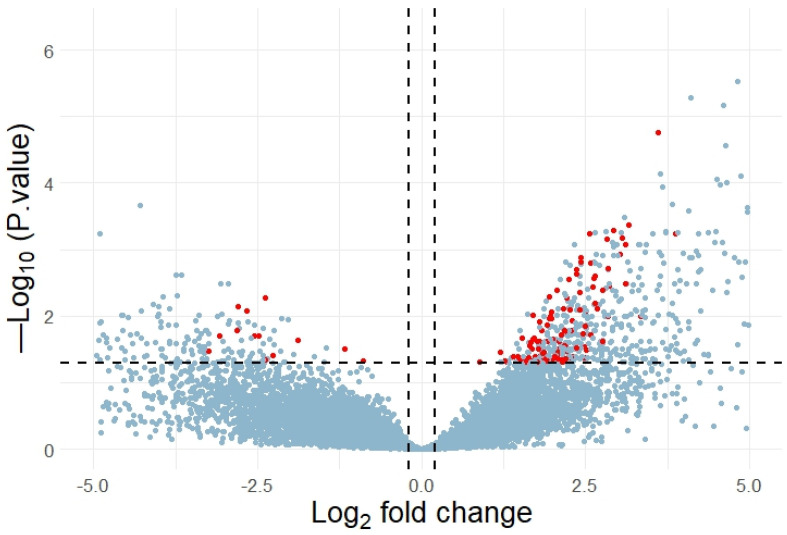
Volcano plot of DEGs. The horizontal axis shows the log_2_ fold change and the vertical axis, –log_10_ (adjusted *p*-value). Significant DEGs with detected rSNPs are colored red.

**Table 1 ijms-25-09297-t001:** Top 20 traits ordered according to the number of GWAS-derived rSNPs.

Trait	Total Number of SNPs per Trait in GWAS Catalog	Number of rSNPs Directly Included in GWAS Catalog	p.adj.	Odds Ratio *
Waist-to-hip ratio adjusted for BMI	3741	36	1.1 × 10^−3^	2.2 (1.5–3.1)
Mean corpuscular volume	2227	31	1.1 × 10^−5^	3.2 (2.2–4.6)
White blood cell count	2456	31	6.1 × 10^−5^	2.9 (2–4.2)
Waist–hip index	2526	28	1.1 × 10^−3^	2.5 (1.7–3.7)
Hip circumference adjusted for BMI	3359	28	4.1 × 10^−2^	1.9 (1.3–2.8)
Platelet count	2609	27	3.9 × 10^−3^	2.4 (1.6–3.5)
Mean corpuscular hemoglobin	2365	25	4.8 × 10^−3^	2.4 (1.6–3.6)
Lymphocyte count	1651	24	9.8 × 10^−5^	3.3 (2.1–5)
Monocyte count	1767	24	2.1 × 10^−4^	3.1 (2–4.7)
Red blood cell count	2487	24	1.9 × 10^−2^	2.2 (1.4–3.3)
Neutrophil count	1559	23	1.1 × 10^−4^	3.4 (2.1–5.1)
Eosinophil count	2100	23	4.8 × 10^−3^	2.5 (1.6–3.8)
Type 2 diabetes	2830	23	7.1 × 10^−2^	1.9 (1.2–2.8)
Red cell distribution width	1732	20	5.5 × 10^−3^	2.6 (1.6–4.1)
Mean platelet volume	1402	16	2.1 × 10^−2^	2.6 (1.5–4.3)
A body shape index	1514	16	3.6 × 10^−2^	2.4 (1.4–3.9)
Monocyte percentage of white cells	738	15	1.9 × 10^−4^	4.6 (2.6–7.7)
Plateletcrit	930	15	1.6 × 10^−3^	3.7 (2–6.1)
Mean spheric corpuscular volume	821	14	1.6 × 10^−3^	3.9 (2.1–6.6)
Appendicular lean mass	1569	14	8.8 × 10^−2^	2 (1.1–3.4)

* 95% confidence interval is parenthesized.

**Table 2 ijms-25-09297-t002:** rSNPs in promoters of the hub genes with AUC > 0.7.

Gene	Regulation	AUC	Rs_id
*NOTCH1*	Up	0.7212	rs951509664, rs3013307, and rs3013306
*H6PD*	Up	0.8072	rs184437520, rs3752547, rs9435144, and rs11121354
*POLR2A*	Up	0.7314	rs4796424, rs57985740, rs41555218, rs144575559, and rs9901161
*NCOR2*	Up	0.7713	rs7960906, rs1006100, rs1199426444, rs191752208, rs1432659465, rs998518300, rs906886068, rs1458070990, rs948418315, rs79830634, rs12426514, rs1316249, rs924583078, rs868110059, and rs1407929149
*PXN*	Up	0.7181	rs7953949 and rs3890165
*FASN*	Up	0.7642	rs7209621 and rs62078751
*SCARB1*	Up	0.7323	rs7305310, rs838884, and rs897715
*GAK*	Up	0.734	rs140032537, rs141564663, rs1403319282, rs182955420, rs3775124 and rs3733352
*CTSD*	Up	0.7238	rs2292963, rs144932926, rs2292962, and rs35640004
*FZR1*	Up	0.7145	rs8100223 and rs8644
*SMG1*	Up	0.7926	rs142606705, rs12929094, and rs560580650
*TP53*	Up	0.7522	rs1800899
*MAN1B1*	Up	0.7762	rs4880199 and rs10870178
*RPL3*	Down	0.766	rs5757613, rs2072872, rs137626, rs2076125, rs143897309, rs969895370, rs84491, rs137627, rs470081, rs754570306, rs6509, rs137620, and rs12484030
*RPS3*	Down	0.7119	rs186612441
*HSP90AB1*	Down	0.8227	rs324131
*RPL11*	Down	0.801	rs3753270, rs111953674, rs878908315, rs558662093, and rs1361739260
*RPS11*	Down	0.7358	rs739349
*RPL5*	Down	0.7411	rs34244251
*RPS20*	Down	0.7159	rs17814456
*RPL13A*	Down	0.7101	rs11539123
*RACK1*	Down	0.8054	rs2287715 and rs111326428
*RPL14*	Down	0.7908	rs62263890 and rs2276869
*RPL12*	Down	0.8285	rs2247310 and rs2247322
*CCT7*	Down	0.8027	rs779122697
*RUVBL1*	Down	0.7451	rs11719546
*TCP1*	Down	0.7292	rs62621403
*PSMA7*	Down	0.8005	rs73307256, rs6089665, and rs3746651
*PSMC5*	Down	0.7863	rs141975038
*ATP5F1A*	Down	0.7092	rs3753069, rs41274316, and rs34907121
*HNRNPK*	Down	0.742	rs1011582290, rs796004, and rs296890
*SKP1*	Down	0.7163	rs56257643
*BCL2*	Down	0.7176	rs4987834 and rs1160274961
*FYN*	Down	0.7438	rs62413757, rs189480003, rs71564104, rs9487736, rs1295051482, rs6568706, rs17072881, and rs1057979
*SSRP1*	Down	0.8262	rs61888886 and rs61888888
*CDC42*	Down	0.7247	rs61778042, rs866829764, rs11801382, rs2255282, rs12038474, and rs16826302

## Data Availability

The original data presented in the study are openly available in Sequence Read Archive (SRA) at https://www.ncbi.nlm.nih.gov/sra, PRJNA1120824 (accessed on 6 June 2024).

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
