# Peer review of "Multi-Omics Analysis Revealed the rSNPs Potentially Involved in T2DM Pathogenic Mechanism and Metformin Response"

_ijms, 2024, doi:10.3390/ijms25179297_

Round 1

Reviewer 1 Report

Comments and Suggestions for Authors

This is quite an interesting study applying multi-omics approaches to the identification of  rSNPs associated with T2D and related phenotypes.

They focused on both ASB (allele-specific binding) of TFs and ASE(allele-specific events) of differential expression of target genes, and demonstrated a considerable number of GWAS-identified SNPs were actually rSNPs. The also found that several interesting gene clusters  were associated with the phenotypes. I appreciate the laborious work by the authors, and have only a few minor comments.

1)    Please remind the readers that the expression profiles were mainly derived from the PBMCs, and that there may be more of significant rSNPs in other specific affected organs, such as pancreatic beta cells, liver, muscle, gut, and even brain.

2)    The rSNPs related to metformin response is interesting, but the caution should be paid, because it depended on how the responsiveness of metformin was defined. It is also important whether the expression profiles between metformin responders and non-responders were consistent with the rSNPs identified in this study.

3)    Despite the above point, it would be interesting to compare the rSNPs set associated with T2D and those associated with metformin response.

4)    I wonder why no rSNPs were significant between control and T2D patients without retinopathy.

Author Response

Thank you for reading our article carefully. We are very grateful for your interest and for comments to our manuscript.

1. Comment. Please remind the readers that the expression profiles were mainly derived from the PBMCs, and that there may be more of significant rSNPs in other specific affected organs, such as pancreatic beta cells, liver, muscle, gut, and even brain.

1. Response. We agree with your comment. The corresponding phrase has been added to the "Discussion" section, page 16, line 388-390.

« Note that obtaining similar data for other specific affected organs, such as pancreatic beta cells, liver, muscle, etc., would significantly expand the rSNP panel».

2.1. Comment. The rSNPs related to metformin response is interesting, but the caution should be paid, because it depended on how the responsiveness of metformin was defined.

2.1. Response. The necessary clarification has been added to the «Materials and Methods» section, page 22, line 666-669.

«According to [78] the patients who had the decrease in glycated haemoglobin (HbA1c) level > 1% or > 20 mg/dl decrease in fasting blood glucose level from baseline after three months of therapy were considered as responders.»

2.2.Comment. It is also important whether the expression profiles between metformin responders and non-responders were consistent with the rSNPs identified in this study.

2.2.Response. Unfortunately, the patient sample size of the RNA-seq studies (10 responders, 10 non-responders) (GSE153315; Vohra M. et al., 2022) does not allow for this analysis.

Vohra, M.; Sharma, A.R.; Mallya, S.; Prabhu, N.B.; Jayaram, P.; Nagri, S.K.; Umakanth, S.; Rai, P.S. Implications of Genetic Variations, Differential Gene Expression, and Allele-Specific Expression on Metformin Response in Drug-Naïve Type 2 Diabetes. J Endocrinol Invest 2022, 46, 1205–1218, doi:10.1007/s40618-022-01989-y.

 3. Comment. Despite the above point, it would be interesting to compare the rSNPs set associated with T2D and those associated with metformin response.

3. Response. Unfortunately, among the 367 rSNPs localized in the promoters of DEGs between metformin responders and non-responders (GSE153315; Vohra M. et al., 2022), we did not find any of the 23 rSNPs in our panel associated with T2DM according to the GWAS data. However, interestingly, among the 3810 rSNPs localized in the promoters of DEGs between controls and T2DM patients with retinopathy (GSE221521; Xiang, Z.-Y. et al., 2023), we found 4 rSNPs associated with T2DM according to the GWAS data. The result is presented in the «Results» section, page 8, line 204-211.

  Vohra, M.; Sharma, A.R.; Mallya, S.; Prabhu, N.B.; Jayaram, P.; Nagri, S.K.; Umakanth, S.; Rai, P.S. Implications of Genetic Variations, Differential Gene Expression, and Allele-Specific Expression on Metformin Response in Drug-Naïve Type 2 Diabetes. J Endocrinol Invest 2022, 46, 1205–1218, doi:10.1007/s40618-022-01989-y.

Xiang, Z.-Y.; Chen, S.-L.; Qin, X.-R.; Lin, S.-L.; Xu, Y.; Lu, L.-N.; Zou, H.-D. Changes and Related Factors of Blood CCN1 Levels in Diabetic Patients. Front Endocrinol (Lausanne) 2023, 14, doi:10.3389/fendo.2023.1131993.

4. Comment. I wonder why no rSNPs were significant between control and T2D patients without retinopathy.

4. Response. We suppose that the absence of DEGs (and correspondingly, rSNPs) between the controls and patients with T2DM may be due to the effect of therapy that patients with T2DM may have received. Unfortunately, the authors of the article [Xiang, Z.-Y. et al., 2023] did not indicate whether the patients took any medication or not. However, given the long duration of T2DM (on average 11 years) in these individuals, it should be assumed that they may have received some. These considerations were given in the «Discussion» section. Please, see the page 17, lines 430-433.

Xiang, Z.-Y.; Chen, S.-L.; Qin, X.-R.; Lin, S.-L.; Xu, Y.; Lu, L.-N.; Zou, H.-D. Changes and Related Factors of Blood CCN1 Levels in Diabetic Patients. Front Endocrinol (Lausanne) 2023, 14, doi:10.3389/fendo.2023.1131993.

Reviewer 2 Report

Comments and Suggestions for Authors

This study aimed to identify and assess the functional SNPs role in the development of type 2 diabetes mellitus and\or their effect on individual response to antihyperglycemic medication with metformin. The authors applied bioinformatics approach to identify the regulatory SNPs (rSNPs) associated with allele-asymmetric binding and expression events in paired ChIP-seq and RNA-seq data for peripheral blood mononuclear cells (PBMCs) of nine healthy individuals. Then the rSNP results were further anlyzyed using public data from GWAS and GTEx. 

(1) For SNP study, the real sample size is relatively small.

(2) The RNA expression and SNP results were derived different study population , that lacks of correponding correlation. 

Author Response

Thank you for reading our article carefully. We are very grateful for your interest and for comments to our manuscript.

Comments 1. For SNP study, the real sample size is relatively small.

Response 1. In fact, a large sample size is not quite necessary for rSNPs detection based on allele-specific analysis. This analysis does not compare individuals, as GWAS and eQTLs analyses do, but allelic effects at all heterozygous sites within single diploid organism are analyzed. Since the two alleles at each heterozygous site act in the same genetic background and under the same environmental conditions, this allows for a significant reduction in sample size to even a single individual [Harvey C.T., 2015; Degtyareva A.O., 2021]. An increase in sample size is only required to include a larger number of rSNPs in the heterozygous state in the analysis. For example, some calculations show that data on 20 individuals theoretically allow for the detection of ASE or ASB events for 65–70% of SNPs with a population frequency ≥5% [Cavalli M., 2016].

Harvey, C.T.; Moyerbrailean, G.A.; Davis, G.O.; Wen, X.; Luca, F.; Pique-Regi, R. QuASAR: Quantitative allele-specific analysis of reads. Bioinformatics 2015, 31:1235–1242. doi: 10.1093/bioinformatics/btu802.

Degtyareva, A.O.; Antontseva, E. V.; Merkulova, T.I. Regulatory SNPs: Altered Transcription Factor Binding Sites Implicated in Complex Traits and Diseases. Int J Mol Sci 2021, 22, 6454, doi:10.3390/ijms22126454.

Cavalli, M.; Pan, G.; Nord, H.; Wallerman, O.; Wallén Arzt, E.; Berggren, O.; Elvers, I.; Eloranta, M.-L.; Rönnblom, L.; Lindblad Toh; K.; et al. Allele-specific transcription factor binding to common and rare variants associated with disease and gene expression. Hum. Genet 2016, 135:485-497. DOI 10.1007/s00439-016-1654-x.

Comment 2. The RNA expression and SNP results were derived different study population , that lacks of correponding correlation.

Response 2. Recently, using data from different population groups is a widely used approach in modern multi-omics studies. Our work was carried out within this framework. Based on our experimental data, we formed the rSNP panel, and then used the available omics data to characterize this panel and establish the possible role of the detected rSNPs. The obtained data allow us to assume that the found rSNPs may contribute to the development of T2DM and the response to metformin.

Reviewer 3 Report

Comments and Suggestions for Authors

“Multi-omics analysis revealed the rSNPs potentially involved in T2DM pathogenic mechanism and metformin response” is a very interesting article.

The background clearly states the aim of the study.

The methodology section is described well. More details on how the health status of participants was confirmed would be useful. As well as more participants. You should include additional demographic information such as lifestyle factors and clarify the rationale for using healthy controls and their role in comparison with T2DM patients.

The results section is comprehensive but could be broken down into more clear parts.

The conclusion effectively summarizes the study's findings but could be slightly more concise.

Overall, this is a good study. Well-written and with good language.

Comments on the Quality of English Language

Language is fine .

Author Response

Comment. More details on how the health status of participants was confirmed would be useful. You should include additional demographic information such as lifestyle factors

Response. We are very grateful for your interest and for comments to our manuscript. We revised the manuscript in accordance with your advice, so some anonymized additional information on our donors was added in Supplementary (Table S8) information and cited in the "Materials and Methods " section, page 19, line 533.

Comment. As well as more participants.

Response. A large sample size is not necessary for rSNPs detection based on allele-specific event analysis. This analysis compares not individuals, as in the GWAS and eQTLs analyses, but allelic effects at the heterozygous site within diploid organism. Since the two alleles at each heterozygous site act within the same genetic background and under the same environmental conditions, this allows for a significant reduction in sample size to even a single individual [Harvey C.T., 2015; Degtyareva A.O., 2021]. An increase in sample size is only required to include a larger number of rSNPs in the heterozygous state in the analysis. For example, calculations show that data on 20 individuals theoretically allow for the detection of ASE or ASB events for 65–70% of SNPs with a population frequency ≥5% [Cavalli M., 2016].

Harvey, C.T.; Moyerbrailean, G.A.; Davis, G.O.; Wen, X.; Luca, F.; Pique-Regi, R. QuASAR: Quantitative allele-specific analysis of reads. Bioinformatics 2015, 31:1235–1242. doi: 10.1093/bioinformatics/btu802.

Degtyareva, A.O.; Antontseva, E. V.; Merkulova, T.I. Regulatory SNPs: Altered Transcription Factor Binding Sites Implicated in Complex Traits and Diseases. Int J Mol Sci 2021, 22, 6454, doi:10.3390/ijms22126454.

Cavalli, M.; Pan, G.; Nord, H.; Wallerman, O.; Wallén Arzt, E.; Berggren, O.; Elvers, I.; Eloranta, M.-L.; Rönnblom, L.; Lindblad Toh; K.; et al. Allele-specific transcription factor binding to common and rare variants associated with disease and gene expression. Hum. Genet 2016, 135:485-497. DOI 10.1007/s00439-016-1654-x.

Comment. and clarify the rationale for using healthy controls and their role in comparison with T2DM patients.

Response. The choice of PBMCs from healthy individuals for our study was motivated by the consideration that in patients, there is a very likely increase in the incidence of homozygotes at trait-associated polymorphic sites, which does not allow the detection of allele-asymmetric events at these positions.

Comment. The results section is comprehensive but could be broken down into more clear parts.

Response. According to you comment, we inserted several subheadings into the chapter 2.3 to make it more clear. Namely:

2.3.1 Search for differentially expressed genes related to T2DM and harbouring rSNPs within promotors. Page 7, lines 188-189.

2.3.2 Identification of hub genes and analysis of key modules using STRING based protein interactions, KEGG and GO enrichment. Page 8, lines 217-218.

2.3.3 Selecting important regulators from PPI network using ROC analysis. Page 12, line 305.

Comment. The conclusion effectively summarizes the study's findings but could be slightly more concise.

Response. We appreciate your comment, but we think it would be better not to change the Conclusion in order to leave its meaning clear and comprehensive.

Round 2

Reviewer 2 Report

Comments and Suggestions for Authors
The authors did not answer the questions. They  cliaimed that a large sample size is not quite necessary for rSNPs detection based on allele-specific analysis.